# Spontaneously Opening and Closing Macular Holes with Lamellar Hole Epiretinal Proliferation: A Longitudinal Optical Coherence Tomography Analysis

**DOI:** 10.3390/diagnostics15060759

**Published:** 2025-03-18

**Authors:** Omar Moussa, Jedrzej Golebka, Gabriel Gomide, Dvir Koenigstein, Hueyjong Shih, Royce W. S. Chen

**Affiliations:** Department of Ophthalmology, Edward S. Harkness Eye Institute, Columbia University Irving Medical Center, New York-Presbyterian Hospital, New York, NY 10032, USA

**Keywords:** epiretinal membrane, lamellar hole epiretinal proliferation, macular hole, full-thickness macular hole, optical coherence tomography

## Abstract

**Background/Objectives**: Spontaneous macular hole closure is a rare phenomenon, with lamellar hole epiretinal proliferation (LHEP) frequently implicated as a potential mechanism. This study aims to analyze the presence of LHEP in patients with full-thickness macular holes (FTMHs) or lamellar macular holes (LMHs) that closed spontaneously without intervention. **Methods**: A retrospective longitudinal analysis of optical coherence tomography (OCT) scans was conducted for 73 patients diagnosed with FTMH or LMH in a single institution. Patients with documented spontaneous hole closure were further analyzed for the presence of LHEP, other OCT findings, and clinical characteristics. **Results**: Of the 73 patients, eight (11%) exhibited spontaneous closure of their macular holes. LHEP was identified in all cases, regardless of hole type (FTMH or LMH). Associated OCT features on diagnosis included VMT in one eye (13%), partial or complete posterior vitreous detachment in seven eyes (88%) and epiretinal membrane in eight eyes (100%). During hole closure, an outer nuclear layer bridge was noted in six eyes (75%). Various extents of outer retinal recovery were noted following closure. After closure, five holes (63%) remained closed without further intervention, while three (38%) reopened and required surgical intervention. **Conclusions**: Spontaneous macular hole closure is strongly associated with the presence of LHEP, highlighting its potential role in retinal repair mechanisms. While in most patients the spontaneous closure is permanent, surgical intervention may be necessary in cases of hole recurrence.

## 1. Introduction

Macular holes are classified as either partial-thickness lamellar holes (LMHs) or full-thickness macular holes (FTMHs) depending on the extent of the retinal defect [1]. In many cases, macular holes are associated with the presence of an epiretinal membrane (ERM) or vitreomacular traction (VMT), which generate tractional forces on the retina that can ultimately lead to the formation of a hole [2]. Randomized controlled clinical trials have demonstrated a clear benefit for surgical intervention to facilitate hole closure [3,4]. Nonetheless, these trials have also documented a small but significant number of spontaneously closed macular holes. The incidence of spontaneous macular hole closure has been reported to vary between 4 and 11% across different cohorts, varying according to the stage of the hole [5]. The exact mechanism behind spontaneous hole closure remains unknown.

Recent studies have suggested that lamellar-hole-associated epiretinal proliferation (LHEP) may play a role in spontaneous macular hole closure. LHEP, as described by Pang et al. in 2014, is visible on optical coherence tomography (OCT) imaging as an epiretinal tissue with homogenous moderate reflectivity located at the edges of a macular hole [6,7]. It is thought to originate from retinal glial cells in the middle retinal layers of retinal defects. LHEP has been reported in 30.5% of eyes, with LMH and 8.0% eyes with FTMHs [6]. Previous studies proposed that LHEP might represent the retina’s attempt to repair a macular hole by generating a pulling force that brings the hole edges together, counteracting the opposing forces originating from an ERM or VMT [8,9]. However, this hypothesis remains unproven. While some researchers have reported an association between LHEP and spontaneous hole closure [10], others have suggested a possible link between LHEP and hole enlargement [11].

Therefore, this study aims to explore the association between LHEP and spontaneous macular hole closure through longitudinal OCT image analysis of selected cases.

## 2. Materials and Methods

This study was approved by an institutional review board and conducted in accordance with the tenets of the Declaration of Helsinki. In this retrospective case review, the clinical charts of all patients diagnosed with either LMH or FTMH at the Edward S. Harkness Eye Institute/Columbia University Irving Medical Center between 1 November 2017 and 1 November 2018 were reviewed. All patients underwent comprehensive ophthalmologic examination, and their macular holes were evaluated using spectral-domain optical coherence tomography (SD-OCT) (Carl Zeiss Meditec, Dublin, CA, USA). FTMH and LMH were classified based on the SD-OCT criteria outlined in the International Vitreomacular Traction Study [1].

Patients with a history of prior macular surgery or retinal diseases such as diabetic retinopathy, neovascular macular degeneration, or high myopia (>−6 dioptres) were excluded from the study. During the follow-up period, patients with any other ophthalmic conditions that could confound visual acuity outcomes were also excluded from the analysis.

We retrospectively analyzed SD-OCT cube and 5-line raster scans from 73 patients diagnosed with either LMHs or FTMHs. The SD-OCT scans from each patient visit were examined to assess the development or spontaneous resolution of macular hole. Of the 73 patients, 8 were identified as having spontaneous macular hole closure. Their OCT scans were further analyzed for the presence or absence of LHEP. Additional features analyzed and measured on OCT imaging included the presence of ERM, vitreomacular interface disease, cystoid macular oedema (CME), macular hole size (as per International Vitreomacular Study group criteria [1]), and status of the outer retinal structures, such as the outer nuclear layer (ONL), external limiting membrane (ELM), and ellipsoid zone (EZ).

For patients with recurrent macular holes requiring surgical intervention, a standardized surgical approach was followed. This included pars plana vitrectomy, confirmation of a posterior vitreous detachment (PVD), membrane staining with Brilliant Blue dye (DORC, Zuidland, The Netherlands), and peeling of the inner limiting membrane (ILM) and epiretinal membrane. Finally, a non-expansile sulfur hexafluoride (SF_6_) gas tamponade was applied. Postoperatively, patients were instructed to maintain strict face-down positioning for 3 to 5 days. Written informed consent for the publication of their clinical details was obtained from all study participants.

## 3. Results

### 3.1. Case Reports

#### 3.1.1. Patient 1, Left Eye

A 71-year-old myopic female presented with a history of primary open-angle glaucoma, bilateral laser-assisted in situ keratomileusis (LASIK), two Baerveldt tube implants OS, and a prior FTMH OD that has been treated with vitrectomy, membrane peeling, and gas tamponade in 07/2017, but with no subsequent improvement in visual acuity.

During routine glaucoma follow-up on 4/10/2018, OCT imaging revealed a new FTMH OS, associated with mild VMT, LHEP with ERM, and CME. Her BCVA at diagnosis was 20/100. Notably, at her retina clinic visit five days later, OCT imaging showed a reduction in the size of the macular hole, formation of a tissue bridge at the level of the ONL, and partial resolution of the CME. Given her rapid improvement and complex ocular history, a decision was made to closely observe the hole.

By 31/10/2018, complete closure of the hole was observed on OCT imaging, accompanied by persistent VMT, full resolution of CME, a hyperreflective vertical line at the fovea, and partial repair of the ELM and EZ. By 01/2019, the hyperreflective line has disappeared, but a defect in the ELM and EZ persisted. As a result, her BCVA remained stable at 20/100 (Figure A1).

#### 3.1.2. Patient 2, Left Eye

A 77-year-old male with a history of primary open-angle glaucoma OU and surgically repaired retinal detachment OD was referred to the retina clinic by his glaucoma physician for further retinal evaluation. In 02/2016, OCT imaging and fundus examination of the left eye revealed a PVD with an associated LMH, ERM, and LHEP. At the time of diagnosis, his BCVA was 20/40 OS.

In 10/2016, the patient developed a small FTMH with associated CME, resulting in a decline in BCVA to 20/60 OS. He was initiated on non-steroidal anti-inflammatory eye drops and monitored monthly. Over the following six months, OCT imaging showed gradual closure of the macular hole, resolution of CME, a tissue bridge at the ONL, a vertical hyperreflective foveal line, and partial recovery of the ellipsoid zone (EZ). Despite the persistence of a small subretinal fluid pocket, his BCVA had improved to 20/30 OS by 02/2017 due to ELM and EZ regeneration.

The subretinal fluid pocket remained stable throughout the follow-up period but was still present in 02/2018 when the patient experienced a recurrence of the FTMH. In 04/2018, he underwent a 25-gauge pars plana vitrectomy, inner limiting membrane (ILM) peeling, and tamponade with a 20% sulfur hexafluoride (SF_6_) gas bubble. This intervention resulted in complete resolution of the macular hole and subretinal fluid. Postoperatively, his BCVA stabilized at 20/40 OS (Figure 1).

#### 3.1.3. Patient 3, Left Eye

A 78-year-old female with a history of central serous chorioretinopathy and a macular hole OD, treated with vitrectomy, membrane peel, and gas tamponade in 2012, presented in 11/2017 with an FTMH associated with ERM and LHEP in the left eye, identified on OCT. Throughout subsequent visits in 01/2018 and 04/2018, the macular hole remained stable with progression from partial to complete PVD and with a BCVA of 20/20 −2 OS.

Between 04/2018 and 10/2018, significant changes were observed in the left eye. The tissue bridge traversing the macula began to collapse, while the ellipsoid layer started to reform. By 04/2019, the macular hole had closed and the ellipsoid layer was almost fully reformed. Between 04/2019 and 10/2019, further remodeling of the ellipsoid layer was noted, although a few intraretinal cystic spaces persisted. As of 08/2021, a small area of missing photoreceptors remained, but the vision remained at 20/20 OS (Figure 2).

#### 3.1.4. Patient 4, Left Eye

An 82-year-old female with a history of dry age-related macular degeneration OU, PVD OS, and a macular hole OD that was successfully closed after vitrectomy, membrane peel, and gas tamponade in 2016. In 08/2017, OCT imaging of the left eye revealed an LMH with preserved ellipsoid zone, LHEP, and ERM. Her BCVA at the time was 20/40 +2 OS.

By 12/2017, an FTMH with a small ONL tissue bridge had developed OS. Subsequently, by 06/2018, the macular hole had spontaneously closed, showing a vertical hyperreflective foveal line and recovery of the ellipsoid zone on OCT. Throughout the follow-up period, the patient’s BCVA remained stable at 20/20 OS (Figure A1).

#### 3.1.5. Patient 5, Right Eye

Patient 5 was a 70-year-old myopic male with a history of POAG OU, PVD OU, type 1 diabetes, and hyperlipidemia. The patient was regularly followed up for glaucoma. On 12/2016, his BCVA was 20/40 OD, and OCT imaging revealed a perpendicular hyperreflective line in the fovea, indicative of recent hole closure with subtle LHEP membrane changes.

By 4/2018, his BCVA had decreased to 20/60 OD, and OCT showed an outer retinal hole in the fovea with ellipsoid loss and a small area of bridging tissue in the ONL. One month later, his BCVA had improved to 20/40, and OCT demonstrated hole closure with ellipsoid recovery and vertical hyperreflective changes in the fovea (Figure A1).

#### 3.1.6. Patient 6, Right Eye

A 73-year-old female with a history of narrow-angle glaucoma OU presented to the glaucoma clinic on 17/5/2016 with a BCVA of 20/25 OD. OCT imaging revealed a thick hyperreflective ERM, a single large cystic inner retinal cavity at the fovea, and incomplete PVD OD. On 24/5/2016, she underwent laser peripheral iridotomy OD for glaucoma and ocular hypertension.

By 16/6/2016, her BCVA OD had declined to 20/50, and OCT imaging demonstrated an FTMH associated with ERM, LHEP, and worsening CME. Close monitoring showed partial improvement by 22/7/2016, with OCT revealing bridging tissue over the macular hole. On 27/9/2016, OCT imaging confirmed full resolution of the macular hole, with subretinal hyperreflective material and persistent CME, while her BCVA improved to 20/40.

Subsequently, the patient underwent pars plana vitrectomy, epiretinal membrane peel, gas tamponade, phacoemulsification, and IOL insertion on 26/10/2016. Postoperatively, her BCVA improved to 20/20 (Figure A1).

#### 3.1.7. Patient 7, Right Eye

A 72-year-old female with a history of birdshot chorioretinopathy OU, diabetes, hyperlipidemia, and venous thrombosis presented for evaluation. On 7/2016, her BCVA OD was 20/40, and OCT revealed a small LMH with ERM and LHEP. By 10/2016, OCT demonstrated spontaneous hole resolution with a vertical hyperreflective line over the fovea, though ellipsoid loss was noted. Her BCVA improved to 20/30 during this time.

However, on 12/2016, her BCVA dropped to 20/50, and OCT imaging revealed an FTMH larger than the previously documented LMH, with a tissue bridge at the ONL level. Subsequent visits showed progressive enlargement of the macular hole without spontaneous closure.

In early 2017, the patient underwent pars plana vitrectomy with membrane peel and gas tamponade, which resulted in macular hole closure. Postoperatively, her BCVA returned to 20/30 OD (Figure A1).

#### 3.1.8. Patient 8, Right Eye

A 73-year-old female with a history of PVD OU and breast cancer presented for evaluation in 11/2014, with BCVA 20/40 OD and OCT showing an ERM and LHEP with an LMH in the fovea affecting the outer retinal layers. By 11/2015, OCT demonstrated enlargement and remodeling of the macular hole with EZ disruption in the fovea, though her BCVA remained stable.

On 12/2017, OCT imaging showed partial recovery of the EZ and spontaneous resolution of the macular hole. However, by 4/2018, a recurrent large FTMH with associated CME was noted on OCT, and her BCVA OD had dropped to 20/150.

Subsequently, the patient underwent pars plana vitrectomy with membrane peel and gas tamponade OD, which resulted in macular hole closure. Postoperatively, her BCVA improved significantly to 20/25 (Figure 3).

### 3.2. Analysis of Results

During the study period, 73 patients with a full-thickness or lamellar macular holes were identified in our department, of whom 8 (11%) had a documented spontaneous hole closure. Their relevant demographics and medical information are summarized in Table 1.

The mean age of these patients was 75 yeas (range: 70–82 years). At the time of hole diagnosis, the presenting BCVA ranged from 20/20 to 20/100, with a median of 20/45. Following hole closure, BCVA also ranged from 20/20 to 20/100, but the median improved to 20/35.

Details regarding specific OCT findings from the follow-up period are presented in Table 2. Among the eight patients included in the study, three were diagnosed with an LMH and five with an FTMH. The average hole width was 162 µm for FTMHs (aperture width) and 253 µm for LMHs (opening width). The average time to hole closure was 20 weeks (range: 4–73 weeks). Prior to diagnosis, PVD was present in seven eyes (88%), with complete PVD in five eyes and partial PVD in two. In the remaining patient, vitreomacular traction was observed at the time of diagnosis. ERM was noted in eight eyes (100%), including eyes that exhibited either partial or complete PVD and the eye with VMT. At the time of diagnosis OCT, LHEP was present in all cases, with additional CME in four eyes (50%).

Throughout the follow-up, five patients (63%) achieved lasting spontaneous hole closure without surgical intervention. However, three patients (38%) experienced reopening of the macular hole after initial closure. On average, the re-opening had occurred by 6 months (range: 2–12 months) after closure. Due to the recurrent nature of the hole, a surgical intervention was undertaken. In all cases, surgery ultimately resulted in lasting hole closure.

During the process of macular hole closure, OCT examinations revealed the formation of tissue localized to the ONL in six patients (75%) including all five FTMH and one of three LMH patients. An example of it is shown on Figure 4. This ONL bridge was thought to represent an additional retinal attempt promoting hole closure.

After hole closure, ONL integrity was restored in all eight patients, although continuous uninterrupted foveal ELM was observed in only five eyes (63%). Additionally, EZ regeneration was noted on OCT images in only four patients (50%). In four eyes, complete regeneration of all three outer retinal layers occurred and was correlated with BCVA improvement in all cases. In the remaining four eyes, partial regeneration led to BCVA stabilization at the initial diagnosis level in three eyes and a small BCVA improvement in the one remaining eye.

## 4. Discussion

Spontaneous macular hole closure is a rare and poorly understood phenomenon. Several mechanisms for the spontaneous closure of idiopathic macular holes have been proposed, including the formation of LHEP, ONL tissue bridge, and changes in the vitreomacular interface. In this case series, we present eight patients with macular holes that spontaneously closed without ophthalmological intervention.

The innate retinal repair mechanisms described in the literature are thought to originate from the retinal Muller cells. Those responses have been described as LHEP and ONL bridging [12,13,14]. In our study, all cases of spontaneous hole closure were associated with the presence of LHEP, consistent with findings from previous research [15,16]. LHEP was observed in both LMH and FTMH eyes. Notably, studies have demonstrated that while ERM and LHEP are usually both present at the epiretinal space around the hole, they are distinct entities at a cellular level [9]. LHEP exhibits immunoreactivity for glial and hyalocyte cell markers and contains native vitreous collagen, in contrast to the myofibroblast-dominated composition of ERMs [9,17]. These findings support the hypothesis that LHEP plays a role in macular hole repair. However, the exact mechanism by which LHEP facilitates closure remains unclear. It is postulated that LHEP may counteract tractional forces from ERMs or VMT, promoting a favorable environment for closure [6].

Formation of an ONL bridge was observed during the closure process in six eyes, including all five eyes with FTMHs and one of three eyes with LMHs. This mechanism has been previously described [18,19] and is thought to originate from growing projections of Muller cells [20]. Muller-cell-driven annular contraction of outer retinal tissues likely draws the edges of the hole together, facilitating structural repair [21]. However, this centripetal movement is believed to be limited, restricting spontaneous closure to smaller holes only [19].

Therefore, the cases described in our study are examples in which the balance of forces exerted by LHEP and ONL bridge vs. ERM favors spontaneous hole closure, not requiring any intervention. However, we also note the fine nature of the balance, as evident by the three cases of hole re-opening following complete closure.

Based on our findings, we propose that LHEP formation and ONL bridging are the primary mechanisms responsible for spontaneous macular hole closure. The role of vitreomacular adhesion release or complete PVD in facilitating closure remains uncertain. While some studies have suggested their importance in spontaneous hole closure [22,23], others have demonstrated that neither VMT nor PVD release is necessary for closure [14,24]. Our findings support this notion, as evidenced by Patient 1, whose macular hole closed without VMT release. This suggests that although vitreous detachment may contribute to favorable conditions, it is not essential for spontaneous hole closure.

In our study, guided by OCT imaging, spontaneous macular hole closure resulted in retinal structural recovery in all cases, although to different degrees. While ONL reconstruction was observed in all cases, complete ELM and EZ regeneration occurred in a respectively smaller subsets of eyes. Previous studies have suggested that the Muller cell response to retinal injury may promote potential retinal cell regeneration including photoreceptors [25,26]. As expected, EZ regeneration led to improvement of BCVA in all cases. Persistent defects in EZ have been reported in previous studies, with ultra-high-resolution OCT revealing these defects to be relatively small and confined to the photoreceptor layer [24].

This study has several limitations. First, the small sample size precludes robust statistical analysis. The rarity of spontaneous macular hole closure, the relatively mild impact of these holes on visual acuity, and potential loss to follow-up of patients with improving symptoms may explain the limited number of cases. Nevertheless, this study represents one of the largest series of LHEP-associated spontaneous macular hole closure cases reported to date. Second, the uneven and irregular follow-up of patients limited our ability to perform a longitudinal analysis of OCT changes over time.

In conclusion, spontaneous macular hole closure is a rare phenomenon appearing to be associated with distinct repair mechanisms originating from within the retina. Our findings provide further evidence strengthening the potential role of LHEP in promoting macular hole closure. In most cases presented in our report, the LHEP-associated spontaneous hole closure was permanent and associated with visual acuity improvement. Further large case–control studies are needed for robust statistical analysis determining the significance of LHEP and other features assessed in our study in spontaneous macular hole closure.

## Figures and Tables

**Figure 1 diagnostics-15-00759-f001:**
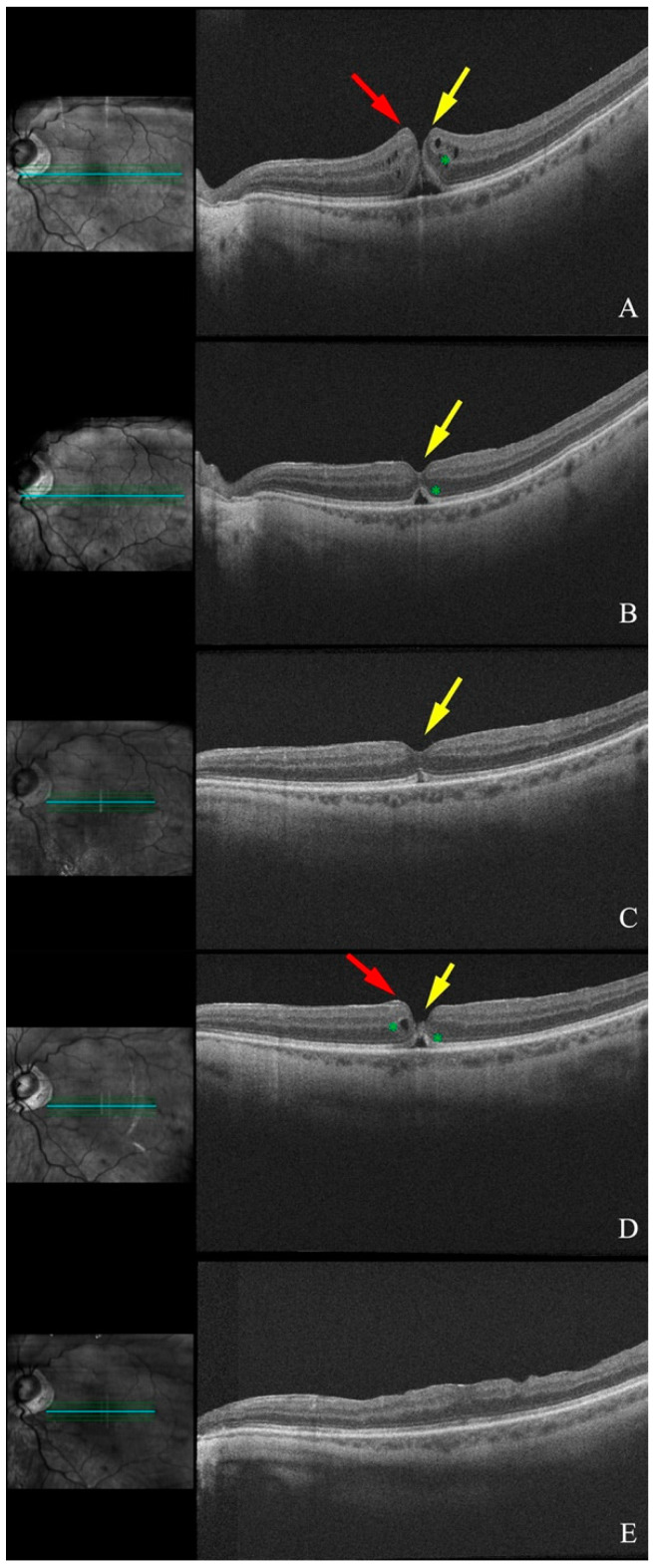
Serial optical coherence tomography (OCT) scans of the left eye (Patient 2): (**A**) Small full-thickness macular hole (FTMH) (yellow arrow) with a few cystoid spaces (green asterisk) in the retina and lamellar hole epiretinal proliferation (LHEP) (red arrow). (**B**) Resolution of the cystoid macular edema and closure of the FTMH (yellow arrow) with a small sub-foveal fluid pocket (green asterisk). (**C**) Resolution of the subretinal fluid and recovery of the ellipsoid (yellow arrow). (**D**) Reopening of the macular hole (yellow arrow) with LHEP (red arrow), cystoid macular edema, and subfoveal fluid (green asterisks). (**E**) Postoperative OCT following par plana vitrectomy (PPV), membrane peel, and gas injection showing resolution of the macular hole with recovery of ellipsoid layer.

**Figure 2 diagnostics-15-00759-f002:**
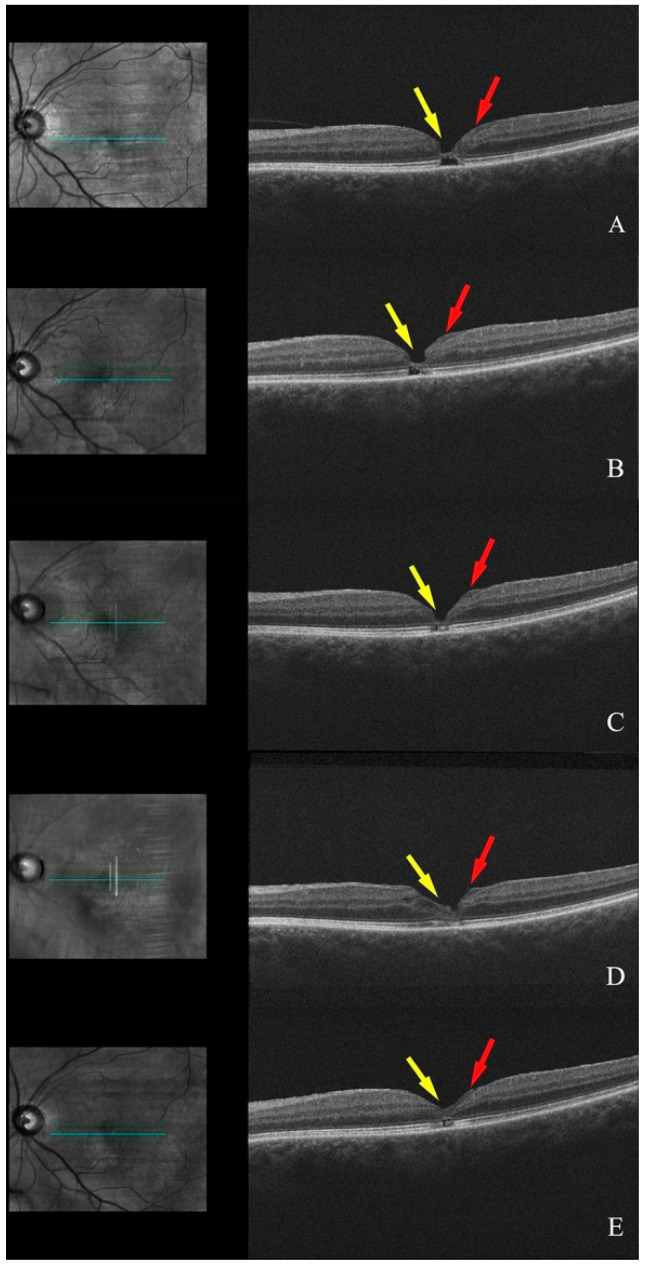
Serial OCT scans of the left eye (patient 3): (**A**) Full-thickness macular hole with bridging tissue (yellow arrow) and LHEP (red arrow). (**B**) Minimal remodeling of the macular hole with more tissue regeneration. (**C**,**D**) Near-complete resolution of the macular hole with partial recovery of the ellipsoid layer. (**E**) Full resolution of the macular hole with small foveal loss of ellipsoid layer and remodeling of the LHEP material to cover the inner foveal surface.

**Figure 3 diagnostics-15-00759-f003:**
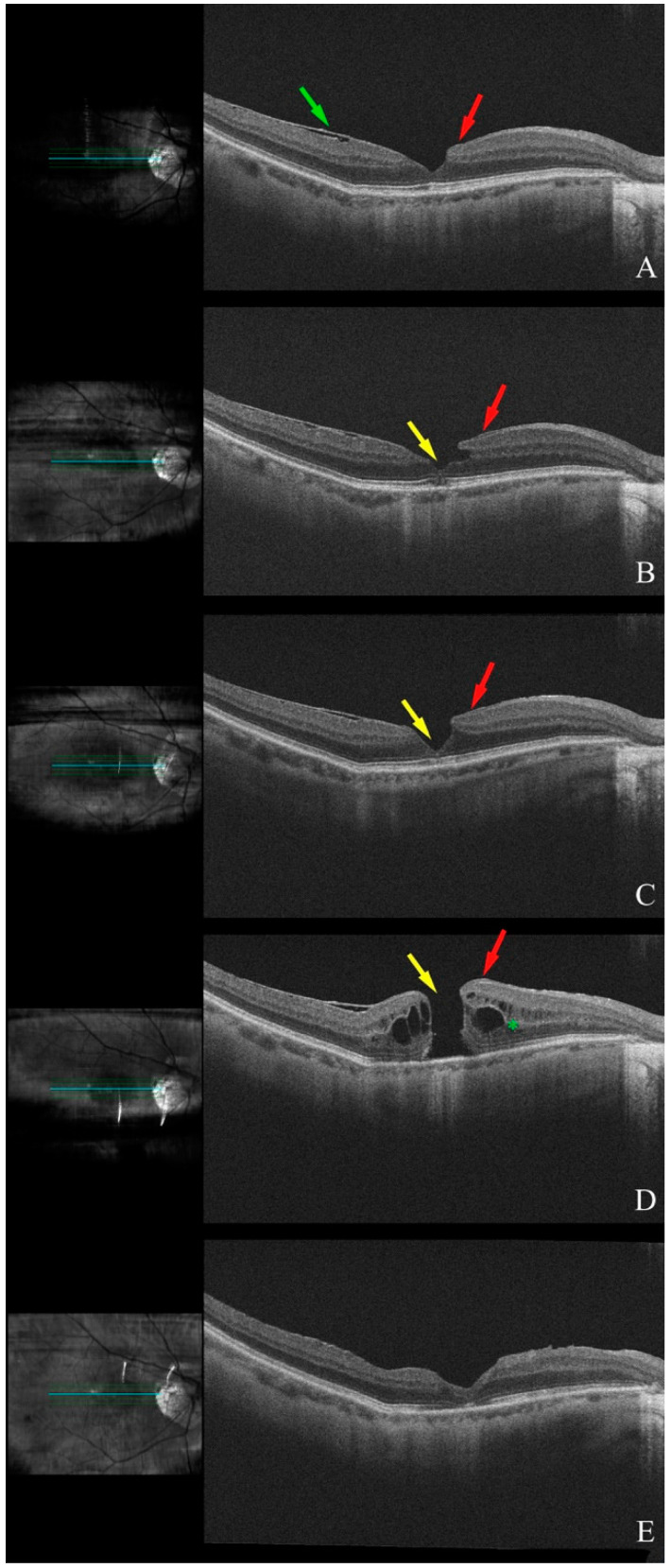
Serial OCT scans of the right eye (Patient 8): (**A**) LHEP (red arrow) and epiretinal membrane (ERM) (green arrow). (**B**) Small macular hole (yellow arrow) developed with remodeling of the retina nasal to the fovea from LHEP changes (red arrow). (**C**) Resolution of the macular hole (yellow arrow) with ellipsoid recovery. (**D**) Large FTMH (yellow arrow) with cystoid spaces in the macula (green asterisk). (**E**) Postoperative OCT following PPV, membrane peel, and gas injection showing resolution of the macular hole with recovery of ellipsoid layer.

**Figure 4 diagnostics-15-00759-f004:**
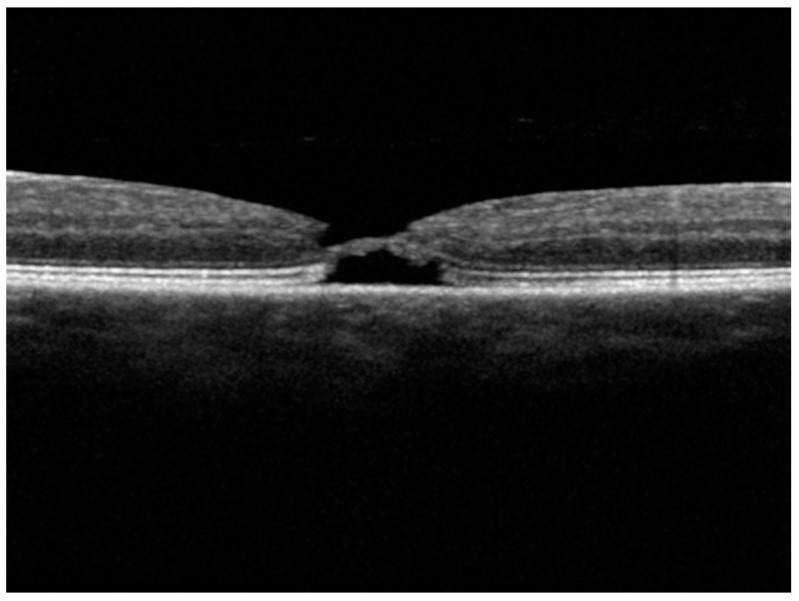
An example of an ONL tissue bridge diagnosed during hole closure in Patient 1.

**Table 1 diagnostics-15-00759-t001:** Demographics and visual acuity and ophthalmological history information for patients with LHEP-associated spontaneous macular hole closure.

Patient	Sex	Age	Eye	Initial BCVA	Final BCVA ^1^	Lens (Phakic, Pseudophakic)	Ocular History
1	F	71	OS	20/100	20/100	Pseudophakic	POAG, myopia, FTMH OD
2	M	77	OS	20/60	20/30	Pseudophakic	Acute anterior uveitis, POAG, retinal detachment OD
3	F	78	OS	20/20	20/20	Phakic	Cataract, CSR, FTMH OD
4	F	82	OS	20/30	20/20	Phakic	Cataract, dry AMD, FTMH OD
5	M	70	OD	20/60	20/40	Pseudophakic	POAG, myopia
6	F	73	OD	20/50	20/40	Phakic	Narrow angle glaucoma
7	F	72	OD	20/40	20/30	Pseudophakic	Birdshot chorioretinopathy
8	F	73	OD	20/40	20/40	Pseudophakic	None

^1^ as noted upon hole closure. BCVA = best corrected visual acuity, POAG = primary open angle glaucoma, FTMH = full-thickness macular hole, CSR = central serous retinopathy, AMD = age-related macular degeneration, OD = right eye, OS = left eye.

**Table 2 diagnostics-15-00759-t002:** Specific longitudinal OCT findings in patients with LHEP-associated spontaneous macular hole closure.

Patient	PVD Before Diagnosis	PVD After Diagnosis	VMT	ERM	LHEP	CME	Hole Type	Hole Width ^1^ (µm)	Closing Time (Weeks)	Time to Recurrence (Months)	ONL Bridge	ONL Integrity Reconstruction	ELM Integrity Reconstruction	EZ Integrity Reconstruction
1	No	No	Yes	Yes	Yes	Yes	FTMH	30	4	-	Yes	Yes	No	No
2	Complete	-	No	Yes	Yes	Yes	FTMH	50	24	12	Yes	Yes	Yes	Yes
3	Partial	Complete	No	Yes	Yes	No	FTMH	270	73	-	Yes	Yes	No	No
4	Complete	-	No	Yes	Yes	No	FTMH	230	24	-	Yes	Yes	Yes	Yes
5	Complete	-	No	Yes	Yes	No	LMH	180	5	-	Yes	Yes	Yes	Yes
6	Partial	Complete	No	Yes	Yes	Yes	FTMH	230	14	-	Yes	Yes	Yes	Yes
7	Complete	-	No	Yes	Yes	No	LMH	210	12	2	No	Yes	Yes	No
8	Complete	-	No	Yes	Yes	Yes	LMH	370	4	4	No	Yes	No	No

^1^ Hole width as aperture width for FTMH and opening width for LMH. FTMH = full-thickness macular hole, LMH = lamellar macular hole, PVD = posterior vitreous detachment, VMT = vitreomacular traction, ERM = epiretinal membrane, LHEP = lamellar hole epiretinal proliferation, CME = cystoid macular edema, ONL = outer nuclear layer, ELM = external limiting membrane, EZ = ellipsoid zone.

## Data Availability

The original contributions presented in the study are included in the article, further inquiries can be directed to the corresponding author. The raw data supporting the conclusions of this article will be made available by the authors on request.

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
