# Peer review of "Spontaneously Opening and Closing Macular Holes with Lamellar Hole Epiretinal Proliferation: A Longitudinal Optical Coherence Tomography Analysis"

_diagnostics, 2025, doi:10.3390/diagnostics15060759_

Round 1

Reviewer 1 Report

Comments and Suggestions for Authors

This study presents a retrospective analysis evaluating the relationship between spontaneous macular hole closure and lamellar hole-related epiretinal proliferation (LHEP). The topic is clinically significant and may contribute to the understanding of the mechanisms of spontaneous closure. However, there is a serious flaw in the methodology of the study:
The possibility that patients requiring surgical intervention (pars plana vitrectomy, PPV) were included in the spontaneous closure group may invalidate the study's basic hypothesis.

Until this flaw is corrected, the validity of the results should be questioned.

The article does not clearly state that patients classified as having spontaneous closure actually recovered without surgical intervention.

In particular, the following cases are problematic in terms of methodology:

Patient 2, Patient 6, and Patient 8:
They were initially reported to have spontaneous closure,
But were later reopened and treated with pars plana vitrectomy (PPV).

If surgical intervention was required, these patients could no longer be considered as having “spontaneous closure.”

The definition of spontaneous closure should be clearly clarified:
"True spontaneous closure" → Cases that do not require any surgical intervention
"Temporary spontaneous closure" → Cases that initially closed but then opened and required PPV
2. Results and Statistical Problems
The calculated "spontaneous closure" rate (11%) may be incorrect.

The article currently states that 8 out of 73 patients showed spontaneous closure. However, if those requiring PPV are excluded from these 8 patients, the true spontaneous closure rate is likely to be much lower.

Patients who underwent PPV should be excluded from the analysis, and the statistics should be recalculated.
The "true spontaneous closure rate" should be clearly stated.
The closure time and the dynamics of reopening of the reopened holes should be detailed.
3. Corrections Needed in Tables and Figures
Tables 1 and 2 do not clearly separate patients requiring surgery from those who indeed closed spontaneously.
An additional table should be added:
"True spontaneous closure" (those that do not require surgery)
"Transient spontaneous closure" (those that reopen and require PPV)
The figure legends should be made clearer and it should be stated which stages are spontaneous and which ones require surgery.

4. Corrections Needed in the Discussion Section
The article gives the impression that spontaneous closure is permanent, but the data show otherwise.
38% of patients had reopening after spontaneous closure and these patients required surgery.
It is not clearly stated how long spontaneous closure lasts.
Whether spontaneous closure is permanent or not should be analyzed more clearly.
The authors may examine in more detail the factors that contribute to the success of spontaneous closure (macular hole width, OCT findings, etc.).

Author Response

Comment 1: The article currently states that 8 out of 73 patients showed spontaneous closure. However, if those requiring PPV are excluded from these 8 patients, the true spontaneous closure rate is likely to be much lower.Patients who underwent PPV should be excluded from the analysis, and the statistics should be recalculated.”

Response 1: We acknowledge this comment and the correct observation that, in 3 cases presented in our paper, a macular hole re-opens following initial spontaneous complete closure. Following hole re-opening a decision to proceed with surgical intervention was made due to recurrent nature of the hole. Nevertheless, we still believe that in all 8 cases a "spontaneous macular hole closure" has occurred as no intervention was conducted. We recognise the need to highlight the 3 aforementioned cases better throughout the paper, although we prefer not to separate them from the rest. We therefore have made the following changes to our manuscript:

1) In the Results section - average time to hole re-opening is added (line 268, "On average, the re-opening occurred by 6 months after closure."); paragraph concerning hole re-opening is changed to better reflect the two groups described by the reviewer (lines 266-270; "Throughout follow-up, 5 patients (63%) achieved lasting spontaneous hole closure without surgical intervention. However, 3 patients (38%) experienced reopening of the macular hole after initial closure. On average, the re-opening occurred by 6 months after closure. Due to recurrent nature of the hole a surgical intervention was undertaken. In all cases, surgery ultimately resulted in lasting hole closure.")

2) In the Discussion section - further answer to comments where we aim to describe the balance of tractional forces described in our paper as fragile, as demonstrated by the 3 cases of re-opening (lines 317-320; "Therefore, the cases described in our study are examples in which the balance of forces exerted by LHEP and ONL bridge vs ERM favors spontaneous hole closure, not requiring any intervention. However, we also note the fine nature of the balance, as evident by the three cases of hole re-opening following complete closure." )

Comment 2: "The article currently states that 8 out of 73 patients showed spontaneous closure. However, if those requiring PPV are excluded from these 8 patients, the true spontaneous closure rate is likely to be much lower.Patients who underwent PPV should be excluded from the analysis, and the statistics should be recalculated."

Response 1: As mentioned above, we believe that all of the 8 patients described in this report, belong to the same cohort of patients in whose a spontaneous hole closure occurred. We understand however that following the closure their future courses differ with 5 patients experiencing lasting closure, while 3 patients experienced hole re-opening. This might be analyzed further in the future with a larger study, but for the purpose of this study, we believe it's appropriate and best if the entire cohort is analyzed together.

Comment 3: "Corrections Needed in Tables and Figures. Tables 1 and 2 do not clearly separate patients requiring surgery from those who indeed closed spontaneously."

Response 3: Patients are presented together in the Tables; however for patients who experienced hole re-opening additional information regarding re-opening time is available 

Comment 4: "The article gives the impression that spontaneous closure is permanent, but the data show otherwise."

Response 4: In our analysis we try to describe the process of spontaneous hole closure due to the presence of innate retinal repair mechanisms such as LHEP. However, we do not aim to say the closure is permanent, as it is not following surgical hole closure, following which hole re-opening is a recognised sequelae. In order to explain this better, we have altered our Discussion section and added information which will hopefully help with that (lines 317-320; "Therefore, the cases described in our study are examples in which the balance of forces exerted by LHEP and ONL bridge vs ERM favors spontaneous hole closure, not requiring any intervention. However, we also note the fine nature of the balance, as evident by the three cases of hole re-opening following complete closure.")

Reviewer 2 Report

Comments and Suggestions for Authors

As a vitreoretinal surgeon I am highly interested in the subject of this paper. I think that it should be published and that the hypothesis about the mechanisms of closure is interesting.

However, I believe that it would be mandatory to present the OCT evolution in all cases.

Also, there are some things that seem unclear to me:

line 152 - what is an "inferior FTMH"?

Figure 2 describes a partial thickness macular hole. Was it a FTMH or not? Perhaps a stage 1 MH?

Author Response

Comment 1: However, I believe that it would be mandatory to present the OCT evolution in all cases.

Response 1: Thank you for the comment. We agree with the opinion and will add the imaging of the remaining cases to the Supplement. 

Comment 2: "Also, there are some things that seem unclear to me:

line 152 - what is an "inferior FTMH"?

Figure 2 describes a partial thickness macular hole. Was it a FTMH or not? Perhaps a stage 1 MH?"

Response 2: Thank you very much for noticing this. We have now corrected the mistake in line 152 calling the FTMH "inferior". We also fixed the mistake in Figure 2 where the hole was incorrectly referred to as a partial thickness macular hole. 

Reviewer 3 Report

Comments and Suggestions for Authors

(A) Provide an overview/summary of the manuscript

Spontaneous macular hole closure is rare and linked to lamellar hole epiretinal proliferation (LHEP). This study examined LHEP in 73 patients with full-thickness macular holes (FTMH) or lamellar macular holes (LMH). Among them, 8 (11%) experienced spontaneous closure, all with LHEP present. Initial OCT findings showed vitreomacular traction in 1 eye (13%), partial or complete posterior vitreous detachment in 7 eyes (88%), and epiretinal membranes in all 8 eyes (100%). During closure, an outer nuclear layer bridge developed in 6 eyes (75%), leading to varying outer retinal recovery. Post-closure, 5 holes (63%) remained closed, while 3 (38%) reopened and required surgery. The study showed a significant correlation between spontaneous closure and LHEP, highlighting its role in retinal repair, though surgery might be needed for recurrences.

(B) Introduction and discussion

The authors appropriately highlighted their work's aims, significance, and novelty. The data presented support the conclusions.

(C) Materials and methods

The methods and statistical analyses used are appropriate.

(D) Results

The reliability and validity of the results are high.

(E) Reviewer's comment

This study was well done, and the manuscript is well written.

Author Response

Thank you very much for the positive comments!

Round 2

Reviewer 1 Report

Comments and Suggestions for Authors

Thanks for the implementation of my suggestion

Author Response

Thank you for your suggestions and reviews!

Reviewer 2 Report

Comments and Suggestions for Authors

In  my opinion, the images present for patient number 7 do not doccument the described initial lamellar macular hole. Perhaps you can find another, clearer image. If not, perhaps patient 7 should be removed from the paper.

Author Response

Thank you for the comment. We agree that the image quality for the first OCT for Patient 7 is far from ideal. We have changed it to what we think is a better image, and additionally changed the format of the figure hopefully improving visibility. We reviewed the image and believe that there is a small slanted defect on the first OCT. Let us know if this improves the confidence with the diagnosis. However, if you remain unsure we would be happy to remove the patient from the analysis as suggested.
